# HRBP: Hardware-friendly Regrouping towards Block-wise Pruning for Sparse CNN Training

## Abstract

Recently, pruning at initialization and training a sparse network from scratch (sparse training) become increasingly popular. However, most sparse training literature addresses only the unstructured sparsity, which in practice brings little benefit to the training acceleration on GPU due to the irregularity of non-zero weights. In this paper, we work on sparse training with fine-grained structured sparsity, by extracting a few dense blocks from unstructured sparse weights. For Convolutional Neural networks (CNN), however, the extracted dense blocks will be broken in backpropagation due to the shape transformation of convolution filters implemented by GEMM. Thus, previous block-wise pruning methods can only be used to accelerate the forward pass of sparse CNN training. To this end, we propose Hardware-friendly Regrouping towards Block-based Pruning (HRBP), where the grouping is conducted on the kernel-wise mask. With HRBP, extracted dense blocks are preserved in backpropagation. We further propose HRBP++ to reduce zero kernels by extracting common sparse kernel patterns on all kernels within one block. Extensive experiments on CIFAR-10, CIFAR-100, and ImageNet demonstrate that HRBP (HRBP++) can almost match the accuracy of unstructured sparse training methods while achieving a huge acceleration on hardware.

## 1 Introduction

Convolutional Neural Networks (CNN) have accomplished enormous progress on many computer vision tasks, such as classification, detection, and segmentation. However, most successful models are overparameterized and computationally extensive. The excessive computation usually requires tedious training and makes it difficult to deploy cumbersome models into real-world applications. Network pruning (LeCun et al., 1990; Han et al., 2015a;b; Li et al., 2016), which removes unnecessary weights from the heavy dense model, stands as one of the most effective methods to compress a heavy model into a lightweight counterpart while maintaining its accuracy.

Traditionally, network pruning follows a three-step paradigm: 1) training a dense network to convergence; 2) identifying a subset of weights (sparse network) by pruning unnecessary connections; 3) retraining or finetuning the sparse network to recover accuracy. However, dense training is still inevitable in this paradigm. The recent Lottery Ticket Hypothesis (LTH) (Frankle & Carbin, 2019) suggests that sparse network can be trained from scratch (sparse training) to the same accuracy as its original dense model. Consequently, the tedious dense training is unnecessary. During the training process, The sparse structure (sparse mask) can either be static (Lee et al., 2019; Wang et al., 2020; Tanaka et al., 2020) or dynamic (Mocanu et al., 2018; Evci et al., 2020; Liu et al., 2021).

Most sparse training methods (Lee et al., 2019; Wang et al., 2020; Tanaka et al., 2020; Mocanu et al., 2018; Evci et al., 2020; Liu et al., 2021) explore unstructured sparsity only, where zero weights distribute irregularly. Although unstructured sparsity can maintain accuracy at a high sparsity ratio, it brings little training time reduction on modern hardware because the irregular mask leads to poor data locality and low parallelism (He et al., 2017; Mao et al., 2017; Wen et al., 2016). An alternative approach, structured sparsity (He et al., 2017; Liu et al., 2017), where the entire filter or channel is pruned, is more hardware-friendly and computationally efficient. However, it usually leads to more accuracy drop compared to unstructured pruning.

Recently, fine-grained structured pruning becomes popular, which is a trade-off between structured pruning and unstructured pruning. On the one hand, the N:M sparsity (Zhou et al., 2021; Sun et al., 2021) defines blocks to meet requirements that only $N$ weights are non-zero for every continuous $M$ weights, which allows acceleration in the inference phase on modern hardware. The N: M transposable mask (Hubara et al., 2021) further ensures that both the weight matrix $W$ and its transpose $W^T$ follow the same sparsity pattern. Thus it can accelerate both forward pass and backward pass. However, these methods require specialized hardware, *i.e.*, the sparse tensor cores (Zhu et al., 2019). Moreover, the transpose matrix $W^T$ does not describe the backward of CNN accurately. As shown in Fig. 1, the convolution operation is usually implemented with general matrix multiplication (GEMM) on hardware. In this case, calculating the gradient w.r.t inputs requires rotating each kernel first, then conducting kernel-wise transpose, rather than a simple transpose operation (See Sec. 2.1 for more detail). Thus, the transposable masks may not always achieve the expected acceleration on the backward pass of CNN. On the other hand, the regrouping algorithm (block-wise pruning) (Rumi et al., 2020; Yuan et al., 2021; Chen et al., 2022) finds dense blocks by grouping unstructured sparse weights, which can accelerate sparse training on general hardware. However, as shown in Fig. 2, the extracted blocks in forward pass usually cannot be maintained in backward pass. Thus, these methods cannot accelerate the backpropagation as well.

In this paper, we propose the Hardware-friendly Regrouping towards Block-wise Pruning (HRBP) for sparse CNN training. HRBP performs the regrouping algorithm on the kernel-wise mask. Thus, it has the ability to maintain the same dense blocks at both forward and backward pass. Meanwhile, all blocks extracted by HRBP have the same shape, which can alleviate unbalanced workload issues in many-core graphics processing units (GPUs) (Chen et al., 2010). Furthermore, we propose HRBP++ to reduce the number of zero kernels, where all kernels in one group share the same sparse pattern. Specifically, sparse training with fixed HRBP++ can almost match the accuracy of unstructured pruning methods such as SNIP (Lee et al., 2019) and GraSP (Wang et al., 2020), but brings 1.4x and 1.6x overall training acceleration with $90\%$ and $95\%$ sparsity for ResNet. Our main contributions are summarized as follows:

- We detailed analyze the implementation of CNN's forward and backward pass with GEMM, and find that current fine-grained structured pruning methods cannot guarantee the backward acceleration.

- We propose a novel Hardware-friendly Regrouping Block-wise Pruning (HRBP/HRBP++) algorithm that extracts dense blocks from the non-zero weights, while maintaining the spatial regularity of the blocks from the weight transformation of the backward propagation, therefore accelerating both forward and backward of CNN training.

- We conduct extensive experiments on CIFAR-10/100 and ImageNet-1K and demonstrate that sparse training with HRBP can achieve a better trade-off between accuracy and hardware acceleration.

## 2 PRELIMINARIES

### 2.1 CONVOLUTION OPERATION AND ITS IMPLEMENTATION

The weights of a 2D convolutional layer can be defined by $\mathbf{K} \in \mathbb{R}^{C_O \times C_I \times K_h \times K_w}$, where $C_O$, $C_I$, $K_h$ and $K_w$ are the number of output channels, the number of input channels, kernel height, and kernel width, respectively. In convolution operation, each filter $\mathbf{K}_c$ slides over the input feature map $\mathbf{I} \in \mathbb{R}^{C_I \times H_I \times W_I}$ and computes a weighted sum of the mapped input values at a time, which generates one activation map $\mathbf{O}_c \in \mathbb{R}^{H_O \times W_O}$. Thus, all $C_O$ filters conduct $C_O$ times of convolution operations and produce the output map $\mathbf{O} \in \mathbb{R}^{C_O \times H_O \times W_O}$.

**Forward pass with GEMM.** On hardware, the convolution operation is usually implemented with general matrix-matrix multiplication (GEMM) (Chetlur et al., 2014), where tensor is laid out in the memory in the `NCHW` or the `NHWC` format (See Appendix A for more details). We take the `NCHW` format as an example. As shown in Fig. 1(a), for the input $\mathbf{I}$, the `im2col()` operation flattens each convolution window of the input and stacks them as columns in a matrix. Thus, the 2D input feature map $\mathbf{I}$ is unrolled into an input matrix $\mathbf{X} = \text{im2col}(\mathbf{I}) \in \mathbb{R}^{(C_I K_h K_w) \times (H_O W_O)}$. Meanwhile, $\mathbf{K}$ is reshaped and stored in the weights matrix $\mathbf{W} \in \mathbb{R}^{C_O \times (C_I K_h K_w)}$. To this end, the forward pass is

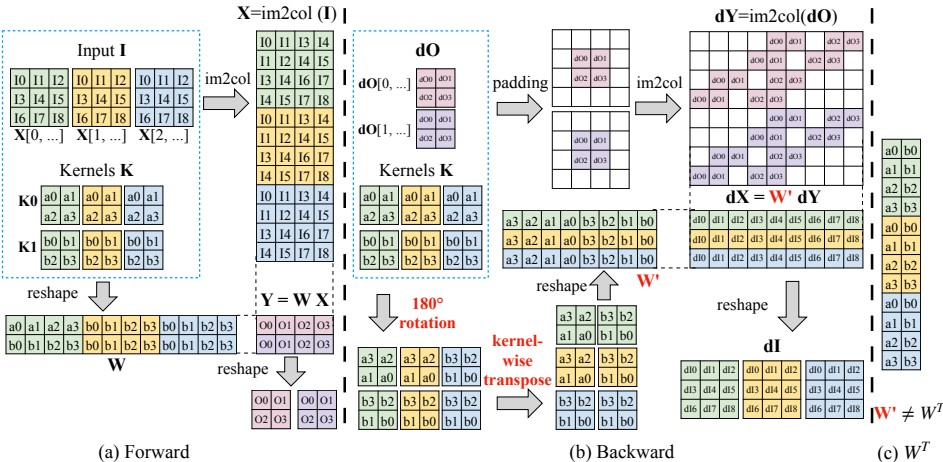

Figure 1: Implementation of forward and backward pass of convolution operation with GEMM in NCHW layout. Different color represents different channels. In the forward pass, the kernels **K** are reshaped to matrix **W**. In the backward pass, each filter is rotated $180°$ firstly, then the kernel-wise transpose is conducted to obtain the new kernel layout **K'**. Then **K'** is reshaped to matrix **W'**, which is different from the transpose **W**$^T$.

calculated by $\mathbf{Y} = \mathbf{WX} \in \mathbb{R}^{C_o \times (H_O W_O)}$. Then the 2D output map **O** is obtained by reshaping **Y**. See Appendix D for a summary of all notations in this paper.

**Backward pass with GEMM.** Given the gradients of the 2D output map $\mathbf{dO} \in \mathbb{R}^{C_O \times H_O \times W_O}$, the backpropagation involves two matrix multiplications. 1) Calculate the gradients w.r.t. the filters **dK**, which is implemented by $\mathbf{dW} = \mathbf{dY} \cdot \mathbf{X}^T$ following $\mathbf{dK} = \texttt{reshape}(\mathbf{dW})$. 2) Calculate the gradients w.r.t the input **dI**, which can be obtained by a full convolution between the kernel **K** and **dO** (LeCun et al., 1989). In detail, as in Fig. 1(b), we conduct padding and `im2col` operation on **dO**, and obtain $\mathbf{dY} = \texttt{im2col}(\mathbf{dO}) \in \mathbb{R}^{(C_O K_h K_w) \times (H_I W_I)}$. Meanwhile, we flip each kernel first vertically and then horizontally (*i.e.*, $180°$ rotation) and perform the kernel-wise transpose to get the new kernel layout **K'**. Then we reshape **K'** to matrix $\mathbf{W'} \in \mathbb{R}^{C_I \times (C_O K_h K_w)}$. Thus, the gradient is calculated by $\mathbf{dX} = \mathbf{W'dY}$. Finally, we reshape **dX** to obtain **dI**.

**Discussion.** For the gradient w.r.t the input, previous works (Hubara et al., 2021) calculate it with $\mathbf{dX} = \mathbf{W^T dY}$ for simplicity, where $\mathbf{W^T} \in \mathbb{R}^{(C_I K_h K_w) \times C_O}$. This simple form is applicable in linear layers. However, it cannot describe CNN accurately in general. As in Fig. 1(c), $\mathbf{W^T}$ is different from the matrix $\mathbf{W'}$. If and only if $K_h = K_w = 1$, they are the same, which downgrades the CNN to a linear layer. Consequently, sparse patterns based on $\mathbf{W^T}$ may not always obtain the expected acceleration on arbitrary CNN backpropagation. Besides, this algorithm has additional requirements on the shape of CNN and the layout in the memory. See Appendix B for more details.

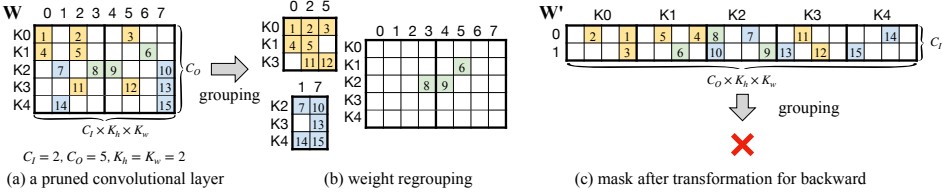

Figure 2: Example of weight regrouping on convolutional layer. Indexed cells are non-zeros. Different color represents different block groups. The extracted dense blocks cannot be kept in the backward pass.

## 2.2 WEIGHT REGROUPING ON UNSTRUCTURED SPARSITY

Given a CNN $f_\theta(\cdot)$ with weights $\theta$, a sparse subnetwork is defined by $f_{\theta \odot \tilde{m}}(\cdot)$, where $\tilde{m} \in \{0, 1\}^{|\theta|}$ is the binary mask and $\odot$ is the element-wise product. In unstructured pruning, zeros distribute irregularly in $\tilde{m}$. Thus, the convolution operation still needs to maintain the entire weight matrix

as a dense network, making it difficult to reduce the computation of unstructured sparse weights on hardware. Even with the help of the dedicated sparse matrix representation technique such as CSR format (Buluç et al., 2009), the unstructured sparsity still expects to have over $85\%$ sparsity ratio to acquire limited acceleration since the irregular weight distribution causes significant computation overhead due to poor data locality (Yuan et al., 2021). The recent weight regrouping (reorganization) (Rumi et al., 2020) can accelerate the unstructured sparse weights on hardware by extracting multiple smaller dense blocks in a large sparse matrix, which can improve the throughput with GEMM.

**Implementation.** Denote the binary mask of the weights of a CNN layer as $m \in \{0, 1\}^{|\mathbf{W}|}$. The regrouping algorithm (Rumi et al., 2020) finds similar rows and columns from the sparse weights matrix $\mathbf{W} \odot m$ and brings them together into several dense blocks. In detail, it firstly clusters $C_O$ rows of $m$ into several groups based on the Jaccard similarity among non-zeros columns. For each group, it then picks out columns with the most non-zeros weights from all $C_I K_h K_w$ columns and generates one dense block. For example, in Fig. 2(a), we can group filter $\mathbf{K}_0$, $\mathbf{K}_1$, and $\mathbf{K}_3$ together, and select columns with at least two non-zero (*i.e.*, $0^{th}$, $2^{th}$, and $5^{th}$ columns), which leads to the dense block with orange color in Fig. 2(b). The remaining sparse weights (cells with green color) are usually discarded (Chen et al., 2022).

**Limitations.** However, the regrouping algorithm cannot be applied to the training directly. Firstly, the extracted dense blocks are fragmentary in backward pass due to the transformation from $\mathbf{W}$ to $\mathbf{W}'$, making the backward acceleration unfeasible. For example, in Fig. 2(c), the cells of the block with orange color in $\mathbf{W} \odot m$ are scattered irregularly in $\mathbf{W}'$. One naive solution is extracting blocks separately in forward and backward passes. However, it is possible that specific weights in the blocks of forward are removed in the blocks of backward, which obstructs the backpropagation of the weight. Besides, the shapes of different dense blocks are arbitrary, which introduces imbalanced memory access and data locality. Thus, it makes the GPU suffer a great workload imbalance.

## 3 METHODOLOGY

### 3.1 HARDWARE-FRIENDLY REGROUPING FOR BLOCK-WISE PRUNING (HRBP)

**Motivation.** The reason for fragmentary blocks in the backward of CNN is that all $C_I K_h K_w$ columns are considered independently. Thus, one dense block can select arbitrary locations on the kernel of different input channels. Ideally, the grouping algorithm should locate identical elements for different input channels within one block. When $K_h = K_w = 1$, the CNN reduces to a linear layer, and the issue of the fragmentary block can be solved naturally. Inspired by this, we propose the HRBP, which extracts dense blocks that can also be kept in CNN backpropagation.

**HRBP.** Different from (Rumi et al., 2020) that conducts regrouping on the mask of weights matrix $m \in \{0, 1\}^{C_O \times (C_I K_h K_w)}$, HRBP conducts regrouping on the kernel-wise mask $m' \in \{0, 1\}^{C_O \times C_I}$. Thus, the entire weights of one kernel $\mathbf{K}_\mathbf{c}^\mathbf{i} \in \mathbb{R}^{K_h K_w}$ are kept or discarded based on $m'$. In detail, we count the number of non-zero cells within each kernel based on $m$, and we keep $rC_I C_O$ kernels with the most number of non-zero cells and discard the remaining kernels, which generates the kernel-wise mask $m'$. Here $r$ is the density ratio of the sparse weights. We then cluster $C_o$ rows of $m'$ into $t$ groups with equal size. For each group, we select $r \cdot C_I$ columns with the most non-zero weights to generate one dense block. Lastly, we discard the remaining elements and refill zero slots in dense blocks. To this end, we have $t$ dense blocks with identical shape $\frac{C_o}{t} \times (rC_I K_h K_w)$. As in Fig. 3(a), the same dense blocks are preserved in backpropagation, making the sparse training with dense blocks applicable in forward and backward passes. See Appendix C for detailed implementation.

**HRBP++.** The vanilla HRBP inevitably introduces too many zero kernels, which may result in layer collapse and a heavy accuracy drop (Tanaka et al., 2020). Thus, we further propose HRBP++ to reduce the number of zero kernels. We define $s$ as the density ratio of each kernel $\mathbf{K}_\mathbf{c}^\mathbf{i}$. In practice, $s$ needs to satisfy $s \geq r \cap s \in \{i/(K_h * K_w)\}_{i=1}^{K_h * K_w}$. Similarly, we select $\frac{r}{s} C_I C_O$ kernels with the most number of non-zero cells to generate $m'$. For each group, we extract a common sparse pattern from all kernels within the group based on $m$. In detail, for each group, we count the number of non-zero weights in each cell of the kernel and choose $sK_h K_w$ cells with the most non-zero weights as a common pattern for the dense block. For example, in Fig. 3(b), one kernel has 4 cells. Given the 6 kernels from the blue group, the number of non-zero weights is $5, 4, 5, 2$ for the 4 cells, respectively. Thus, we select the $1, 2, 3$-th cell as a common pattern and apply it to all kernels

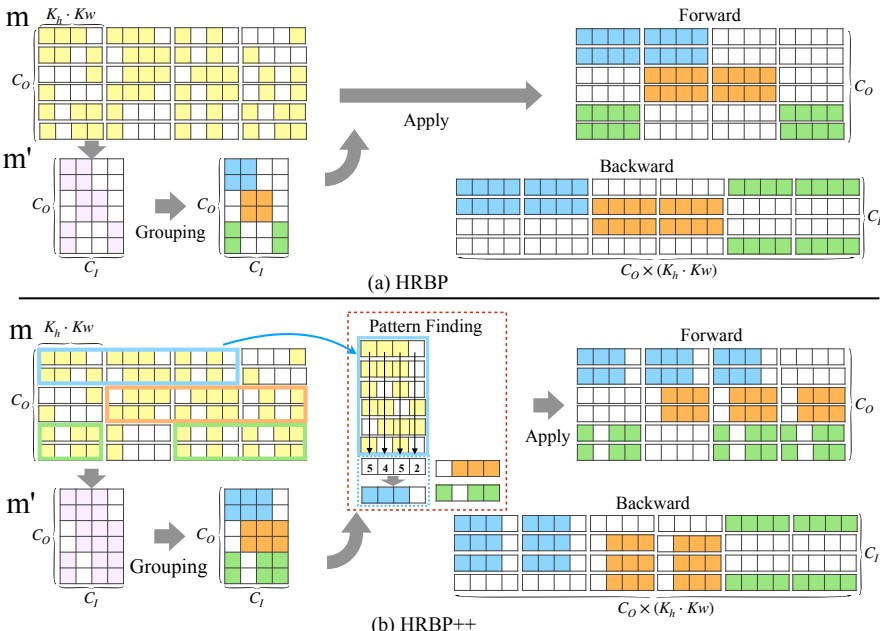

Figure 3: Illustration of HRBP and HRBP++. White cell means zero weights. Given the unstructured mask (matrix with yellow cell), HRBP(++) firstly extracts the kernel-wise mask (matrix with lavender cell) by counting non-zero weights of each kernel. We then group the kernel-wise mask into several equal shape blocks (we have three groups marked with blue, orange, and green). In HRBP, the entire weights of a kernel are kept or discarded. In HRBP++, we extract common patterns of each block and apply it to all kernels within the group.

within the blue group. To this end, HRBP++ allows for zero values within kernels, resulting in fewer entirely discarded kernels for the same overall sparsity ratio.

### 3.2 SPARSE TRAINING WITH HRBP (++)

**HRBP(++)-based Random Mask.** With HRBP(++), we can extract dense blocks from sparse weights and accelerate both forward and backward of sparse training. For simplicity, we apply HRBP(++) to a random pruning mask, which randomly chooses connections to satisfy the sparsity ratio requirements. To this end, we design a strategy to create a random pruning mask in HRBP (++) style, named *HRBP(++)-based Random Mask*. For the sparse ratio of each layer, we follow the Erdős–Rényi-Kernel (ERK) (Evci et al., 2020) where larger layers are allocated with higher sparsity than smaller layers within a network. For HRBP, we randomly split the $C_O$ rows into $t$ equal groups. For each group, we randomly keep $rC_I$ channels and zero out the rest. For HRBP++, we start from a random unstructured mask $m$ and apply the proposed HRBP++ algorithms to slightly modify $m$. Thus, based on the HRBP(++)-based random mask, we can replace the sparse weights matrix with several dense blocks on GPU.

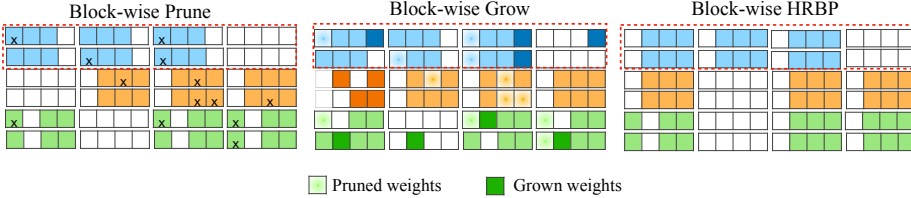

Figure 4: Illustration of Block-wise updating for Dynamic Sparse Training.

**Block-wise updating for Dynamic Sparse Training (DST)** DST starts from a random sparse network. After optimizing several iterations, it prunes a portion of weights based on the pruning criterion and grows new connections according to the grow criterion. Then the new sparse network is trained until the next update. Typically, any location of the entire weight can be pruned or activated

(*i.e.*, grown). However, this default setting is not aligned with HRBP, since updating connections at arbitrary locations destroys the extracted blocks. To this end, we propose block-wise updating for DST with HRBP++, which follows the prune-grow-regroup scheme. As shown in Fig. 4, we maintain the cluster of rows (filters). For each row, we prune $d\%$ weights based on the magnitude and grow back new $d\%$ weights based on its gradient. Then we perform the HRBP++ on the new weights to group the new mask into several dense blocks.

### 3.3 HRBP Hardware Implementation

We implement our method with CUDA and measure the acceleration rate on GPU. In detail, we rewrite the nn.Conv2d module of Pytorch with CUDA to run CNN with sparse weights. The thread is the basic programmable unit that allows GPU programmers to use massive numbers of CUDA cores. CUDA threads are grouped at different levels such as warp, block, and grid. Given the dense blocks found by HRBP/HRBP++, the original convolution computation will be decomposed into several computations corresponding to dense blocks, and these computations will be performed in parallel by threads. The extracted dense blocks from HRBP(++) have the same shape. To this end, in our kernel design, a thread block only processes the output channels of the same group which enables us to have better reuse of the input data (e.g., the input data can be loaded to shared memory which can be access by all the threads inside a thread block). Inside a thread block, each thread is responsible for one portion of the output on $(X, Y)$ dimension which further helps us to achieve good data reuse of the kernel weight (kernel weight can also be put into shared memory).

We also applied tiling across both height and width dimension of the input channel to increase the number of launched thread blocks, which further increases the parallelism level. The tiling size plays a significant role of the performance. A small tiling size can increase the parallelism level, while a big tiling size will provide better reuse of the weight data. Thus, finding a good tile size for a given problem is non-trivial. A simple but powerful method is brute force search, it guarantees to find out the best tiling size for specific hardware. In our design, input data are put into shared memory and can be accessed by all thread block to improve the data reuse. In the meantime, the cost by using brute force search is very low because most of the undesired tiling size candidates are discarded in order to satisfy the usage requirement of the shared memory.

## 4 Experiments

### 4.1 Experimental Setups

**Dataset & Networks.**  We follow the experimental settings in (Wang et al., 2020; Yuan et al., 2021) and conduct experiments on CIFAR-10, CIFAR-100 and ImageNet-1K (Deng et al., 2009). As the common setting in sparse training, we apply the WideResNet-32-2 and VGG-19 (Simonyan & Zisserman, 2014) for CIFAR-10/100 and ResNet-50 (He et al., 2016) for ImageNet.

**Training.**  For CIFAR-10/100, we use a batch size of 128 and train networks with SGD optimizer for 160 epochs by default. The learning rate is set to 0.1 initially and is decayed by a factor of 0.1 at the 80th and 120th. Moreover, we run each experiment 3 times and report the mean value and standard derivation. For ImageNet, we adopt the Pytorch official implementation and train the networks for 100 epoch as (Wang et al., 2020). The learning rate is 0.1 initially and is decayed at 30-th, 60-th, and 90-th epoch with factor 0.1. For sparse training, we set the number of dense blocks $t$ to 8 (Rumi et al., 2020) and the dense ratio of kernel $s$ in HRBP++ to $\frac{4}{9}$ for all $3 \times 3$ kernels. The minimum size of a block $B_1$ is set to 8 (Rumi et al., 2020).

**Hardware**  We evaluate our method on NVIDIA Ampere A100 (108 SMs, 40GB). The versions of CUDA and cuDNN are 11.0.0 and 8.0.4, respectively. For baseline implementations of convolution operations for full sparse weights, we adopt the widely used GEMM-based convolution, i.e., CUDNN_CONVOLUTION_FWD_ALGO_GEMM (Chetlur et al., 2014). We report the overall training time acceleration rate compared with the dense baselines, which includes both forward and backward calculations on hardware.

### 4.2 Sparse Training with HRBP(++)-based Random Mask

**Settings.**  We compare HRBP(++) on static sparse training against random unstructured pruning (RU), random channel-wise pruning (RC) with an ERK ratio, and unstructured pruning with well-defined criteria such as SNIP (Lee et al., 2019) and GraSP (Wang et al., 2020). We also evaluate the

Table 1: Comparison of static sparse training methods on CIFAR-10 and CIFAR-100. The number in brackets is the relative training time speedup compared to a dense model.

| Dataset | CIFAR-10 | | CIFAR-100 | |
|---|---|---|---|---|
| Pruning ratio | 90% | 95% | 90% | 95% |
| ResNet-32 | Dense: 94.80 | | Dense: 74.64 | |
| RU | 92.81±0.19 (1.0×) | 91.38 ±0.04 (1.0×) | 69.48±0.21 (1.0×) | 67.03±0.66 (1.0×) |
| LTH (Frankle & Carbin, 2019) | 92.31 (1.0×) | 91.06 (1.0×) | 68.99 (1.0×) | 65.02 (1.0×) |
| SNIP (Lee et al., 2019) | 92.59 (1.0×) | 91.01 (1.0×) | 68.89 (1.0×) | 65.22 (1.0×) |
| GraSP (Wang et al., 2020) | 92.38 (1.0×) | 91.39 (1.0×) | 69.24 (1.0×) | 66.50 (1.0×) |
| RC | 90.27±0.24 (1.5×) | 87.19±0.48 (1.6×) | 62.71±0.22 (1.5×) | 56.33 ±0.37 (1.6×) |
| Regrouping (Rumi et al., 2020) | 91.63±0.11 (1.1×) | 90.77±0.07 (1.2×) | 66.97±0.18 (1.1×) | 64.35±0.45 (1.2×) |
| HRBP | 91.76±0.17 (1.4×) | 89.80±0.02 (1.6×) | 67.20±0.22 (1.4×) | 63.19±0.73 (1.6×) |
| HRBP++ | 92.30±0.20 (1.4×) | 90.84±0.41 (1.6×) | 69.22±0.50 (1.4×) | 65.94±0.25 (1.6×) |
| VGG-19 | Dense: 94.23 | | Dense: 74.16 | |
| RU | 92.91±0.10 (1.0×) | 91.91±0.13 (1.0×) | 70.39±0.43(1.0×) | 68.63±0.40 (1.0×) |
| LTH (Frankle & Carbin, 2019) | 93.51 (1.0×) | 92.92 (1.0×) | 72.78 (1.0×) | 71.44 (1.0×) |
| SNIP (Lee et al., 2019) | 93.63 (1.0×) | 93.43 (1.0×) | 72.84 (1.0×) | 71.83 (1.0×) |
| GraSP (Wang et al., 2020) | 93.30 (1.0×) | 93.04 (1.0×) | 71.95 (1.0×) | 71.23 (1.0×) |
| RC | 90.84±0.38 (1.7×) | 87.12±0.18 (1.8×) | 59.61±0.52 (1.7×) | 49.31±1.17 (1.8×) |
| Regrouping (Rumi et al., 2020) | 92.81±0.25 (1.2×) | 91.86±0.24 (1.2×) | 70.52±0.36 (1.2×) | 68.60±0.08 (1.2×) |
| HRBP | 92.48±0.19 (1.4×) | 90.81±0.16 (1.9×) | 68.42±0.17 (1.4×) | 64.98±0.47 (1.9×) |
| HRBP++ | 92.88±0.12 (1.4×) | 91.66±0.14 (1.9×) | 70.25±0.29 (1.4×) | 67.89±0.49 (1.9×) |

conventional regrouping method (Rumi et al., 2020) as a baseline, even if it fails to accelerate the backpropagation. Following (Wang et al., 2020), we set the sparsity ratio to 90% and 95%.

**Results.** The results are summarized in Table 1. From the accuracy perspective, HRBP++ achieves similar performance to unstructured pruning methods and much better performance than channel-wise pruning. For example, on CIFAR-10 with ResNet-32, HRBP++ achieves an accuracy of 90.84% at 95% sparsity, which is 3.65% higher than the RC and just 0.17% negligible drop to SNIP (Lee et al., 2019). HRBP, as our baseline with more zero kernels, usually suffers 1% accuracy drop compared to HRBP++, but it still performs much better than RC. From the training time acceleration perspective, unstructured pruning cannot be accelerated due to its irregularity of non-zero weights. However, our HRBP can almost match the acceleration of channel-wise pruning, with the benefit of our optimized convolution implementation. Moreover, HRBP obtains more acceleration compared to the convention regrouping (Rumi et al., 2020), as the latter can only accelerate the forward pass. In summary, our HRBP(++) can successfully achieve comparable accuracy as unstructured pruning and bring training time acceleration on hardware.

**Layer-wise speedup.** We also explore the overall training time acceleration of each convolutional layer. We take VGG-19 with 90% overall sparsity on CIFAR-10 as an example and report the speedup of each layer in Figure 5. At shallow layers with a small sparsity ratio, channel-wise pruning achieves a slightly better acceleration. However, at deeper layers, HRBP achieves comparable or even better results, with at most 7.5x acceleration on `Conv14`. This is consistent with the conclusion in (Rumi et al., 2020) that the grouping algorithm prefers large kernels and large sparsity. Noticeably, cuDNN is only optimized for kernel matrices with a multiple of 32 rows, thus structured pruning with an arbitrary number of channels does not guarantee better acceleration (Rumi et al., 2020).

## 4.3 DYNAMIC SPARSE TRAINING WITH HRBP++

We further explore the dynamic sparse training with HRBP++, which prunes and grows the connections based on the proposed block-wise updating during the training time. We compare DST with HRBP++ to several unstructured DST methods, including DeepR (Bellec et al., 2018), DSR (Mostafa & Wang, 2019), SET (Mocanu et al., 2018), and RigL (Evci et al., 2020). We follow all sparse training hyperparameters in (Evci et al., 2020). As the speedup is similar to static sparse training, we only show the accuracy comparison in Table 2 for simplicity. Noticeably, our HRBP++ can still match the accuracy of unstructured DST methods. Nevertheless, when comparing Table 1 and 2, the advantage of DST from static training in HRBP is not as much as that in unstructured pruning. One potential reason is that the mask updating mechanism of HRBP++ has a smaller mask diversity (Hubara et al., 2021) than the unstructured pruning.

Table 2: Comparison of dynamic sparse training methods on CIFAR-10 and CIFAR-100.

| Network | ResNet-32 | | | | VGG-19 | | | |
|---|---|---|---|---|---|---|---|---|
| Dataset | CIFAR-10 | | CIFAR-100 | | CIFAR-10 | | CIFAR-100 | |
| Pruning ratio | 90% | 95% | 90% | 95% | 90% | 95% | 90% | 95% |
| Deep-R (Bellec et al., 2018) | 91.62 | 89.84 | 66.78 | 63.90 | 90.81 | 89.59 | 66.83 | 63.46 |
| DSR (Mostafa & Wang, 2019) | 92.97 | 91.61 | 69.63 | 68.20 | 93.75 | 93.86 | 72.31 | 71.98 |
| SET (Mocanu et al., 2018) | 92.30 | 90.76 | 69.66 | 67.41 | 92.46 | 91.73 | 72.36 | 69.81 |
| RigL (Evci et al., 2020) | 92.84±0.13 | 92.02±0.29 | 70.98±0.30 | 68.50±0.15 | 93.15±0.09 | 92.30±0.43 | 71.63±0.28 | 69.13±0.46 |
| DST-HRBP++ | 92.72±0.23 | 91.25±0.12 | 69.51±0.52 | 66.41±0.30 | 93.07±0.12 | 91.79±0.18 | 70.49±0.19 | 68.04±0.37 |

Table 3: Static sparse training on ImageNet. The dense ResNet-50 has 75.70% top-1 accuracy.

| Pruning ratio | 60% | | 80% | |
|---|---|---|---|---|
| Accuracy | top-1 | top-5 | top-1 | top-5 |
| SNIP Lee et al. (2019) | 73.95 (1.0×) | 91.97 | 69.67 (1.0×) | 89.24 |
| GraSP Wang et al. (2020) | 74.02 (1.0×) | 91.86 | 72.06 (1.0×) | 90.82 |
| HRBP++ | 74.84 (1.17×) | 92.35 | 70.90 (1.26×) | 89.93 |

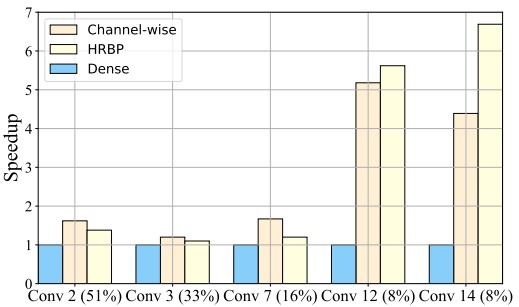

Figure 5: The layer-wise speedup of VGG-19.

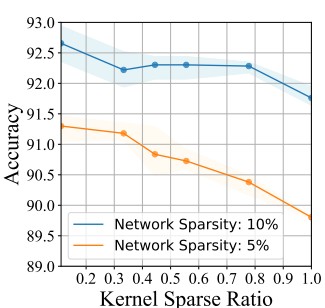

Figure 6: Different kernel dense ratio for HRBP++ with ResNet-32.

## 4.4 Results on ImageNet

We evaluate the static sparse training with HRBP(++) on ImageNet. Follow Lee et al. (2019); Wang et al. (2020), we set the sparse ratio to $60\%$ and $80\%$. As shown in Table 3, HRBP++ can also achieve similar performance as well-designed unstructured pruning methods. Meanwhile, it provides $1.26\times$ and $1.44\times$ training time acceleration on hardware when the pruning ratio is $80\%$ and $90\%$ respectively. To this end, our HRBP++ is still effective on complicated tasks like ImageNet.

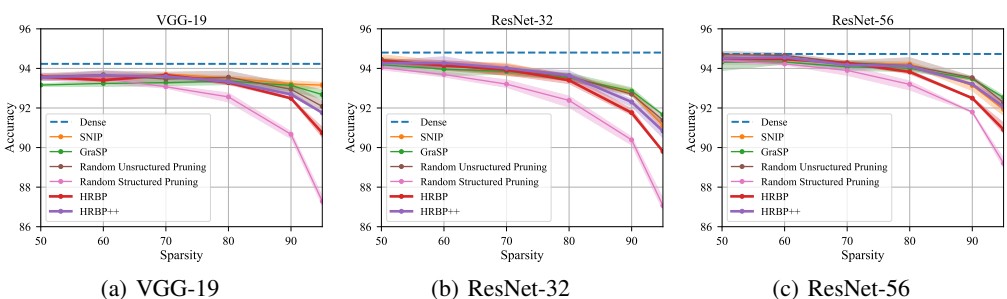

(a) VGG-19

(b) ResNet-32

(c) ResNet-56

Figure 7: The trade-off between sparsity and accuracy. All models are evaluated on CIFAR-10.

## 4.5 Ablation Studies

**Dense Ratio of Kernel Pattern $s$.** We take the ResNet-32 on CIFAR-10 as an example to explore the effect of the kernel sparse ratio $s$ in HRBP++. Typically, a smaller $s$ introduces more non-zero kernels but inevitably reduces the number of non-zero weights within each kernel. As the shape of most kernels is $3 \times 3$, we explore $s$ from $\frac{1}{9}$ to $\frac{9}{9}$. As shown in Fig. 6, a smaller $s$ usually brings slightly better accuracy, especially for networks with large sparsity. This is reasonable since

a smaller $s$ can reduce the number of zero kernels, while many zero kernels can render a network untrainable (Tanaka et al., 2020).

**Sparsity $r$ and deeper nets.** We also vary the sparsity from 50% to 95% to explore the trade-off between sparsity and accuracy. Besides, we also add experiments with deeper nets such as ResNet-56. As in Figure 7, our method can achieve comparable accuracy as unstructured pruning methods at all sparsity levels and all kinds of networks. This suggests the generalization ability of our method. For the acceleration rate on both training and inference, please see Figure 14 in Appendix.

## 5 RELATED WORK

### 5.1 NETWORK PRUNING

Pruning aims to compress overparameterized networks into lightweight ones. Based on the distribution of zero weights, it can be divided into three types: 1) *Unstructured pruning*, where zero weights are distributed at arbitrary locations based on the importance score of each weight. The score can be obtained from magnitude (LeCun et al., 1990; Han et al., 2015a;b; Xu et al., 2021; Zhu & Gupta, 2017), gradient (Molchanov et al., 2017; 2019) or Hessian (LeCun et al., 1990). The unstructured pruning can match the accuracy of dense networks at a high sparsity ratio (more than 90%). However, it is difficult to achieve better hardware speedup due to irregularity. 2) *Structured pruning*, where the weights of entire channels are pruned. Earlier works (Li et al., 2016; He et al., 2017; Wen et al., 2016; Liu et al., 2017) adopt mathematics-oriented regularization-based algorithms to generate sparsity. Other works such as HRank (Lin et al., 2020), SCOP (Tang et al., 2020), DMCP (Guo et al., 2020), MetaPruning (Liu et al., 2019) use complicated rules to generate the sparsity distribution in the channel level. As the sparse model preserves the spatial regularity, the pruned convolution layers can be transformed to a full matrix multiplication with reduced matrix size and accelerate computation on the hardware level. However, structured pruning suffers from significant accuracy loss as one entire activation map can be zero. 3) *fine-grained structured pruning*, which includes block-based pruning and pattern-based pruning. In block-based pruning (Rumi et al., 2020), the unstructured weights are partitioned into several dense blocks. However, all the above works aim to accelerate the inference stage, our work differs substantially from them such that we do not rely on a pre-trained dense model.

### 5.2 SPARSE TRAINING

Sparse training aims to train a sparse network from scratch. Based on the mask updating schemes, it is usually divided into two categories: 1) *Static sparse training*, where the sparse mask is obtained at the early stage of training and is fixed during the course of training. Previous works obtain the sparse mask by random pruning or utilizing some saliency criteria. The SNIP (Lee et al., 2019) uses the gradients of the training loss as connection sensitivity to prune the network at initialization. Later on, many criteria have been proposed, such as gradient flow in GraSP (Wang et al., 2020), synaptic strengths in SynFlow (Tanaka et al., 2020), Fisher information (Sung et al., 2021), etc. 2) *Dynamic sparse training*, which starts from a random sparse mask and dynamically updates connections during training. DST avoids the pruning-at-initialization process that usually involves the full dense model computation. Specifically, SET (Mocanu et al., 2018) update sparse masks by pruning weights that have the least magnitude and grow back the same amount of inactivated weights in a random fashion. RigL (Evci et al., 2020) proposes to update sparse mask by magnitude-based pruning and grow back inactivated weights by their gradients. DSR (Mostafa & Wang, 2019) and STR (Kusupati et al., 2020) design a dynamic reparameterization method that allows weights to be re-distributed across layers by providing a global sparsity allocation dynamics. DeepR (Bellec et al., 2018) combines dynamic sparse parameterization with stochastic parameter updates for training, but it primarily targets small and shallow fully-connected networks. However, these works focus on unstructured sparsity.

## 6 CONCLUSION

We analyze the implementation of CNN with GEMM and point out that current fine-grained structured pruning methods cannot promise backward acceleration. We then propose the HRBP, aiming to accelerate the sparse training at both forward and backward passes by extracting dense blocks from the unstructured mask. Extensive results suggest that sparse training with HRBP(++) can achieve comparable accuracy with unstructured pruning methods. Meanwhile, it brings significant training acceleration. We believe our method can integrate the advantage of both unstructured pruning and structured pruning and make sparse training more hardware-friendly.

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

## A    FORWARD AND BACKWARD PASS IN NCHW AND NHWC LAYOUT

NCHW and NHWC data layout formats are two common types of cuDNN tensors arrangement in memory (Chetlur et al., 2014). These two layouts produce the same shape matrix and the same outputs. The only difference is the order of element in the flattened tensor, as shown below:

**NCHW layout**    As shown in Fig. 8(a), in forward pass, the flattened tensor $\mathbf{W}$ begins with the first input channel (green color), the elements are arranged contiguously in row-major order (*i.e.*, $a0, a1, a2, a3$ with green color for the kernel $\mathbf{K}$ in forward). Then, it continues with second (orange color) and subsequent channels until the elements of all the channels are laid out.

**NHWC layout**    As shown in Fig. 8(c), in forward pass, the flattened tensor $\mathbf{W}$ begins with the first element of the first input channel (*i.e.*, $a0$ with green color for $\mathbf{K}$), then proceed to the first element of the second input channel (*i.e.*, $a0$ with orange color for $\mathbf{K}$), and so on, until the first elements of all the $C$ channels are laid out. Next, select the second element of the first input channel (*i.e.*, $a1$ with green color for $\mathbf{K}$), then proceed to the second element of the second input channel (*i.e.*, $a1$ with orange color for $\mathbf{K}$), and so on, until the second element of all the channels are laid out.

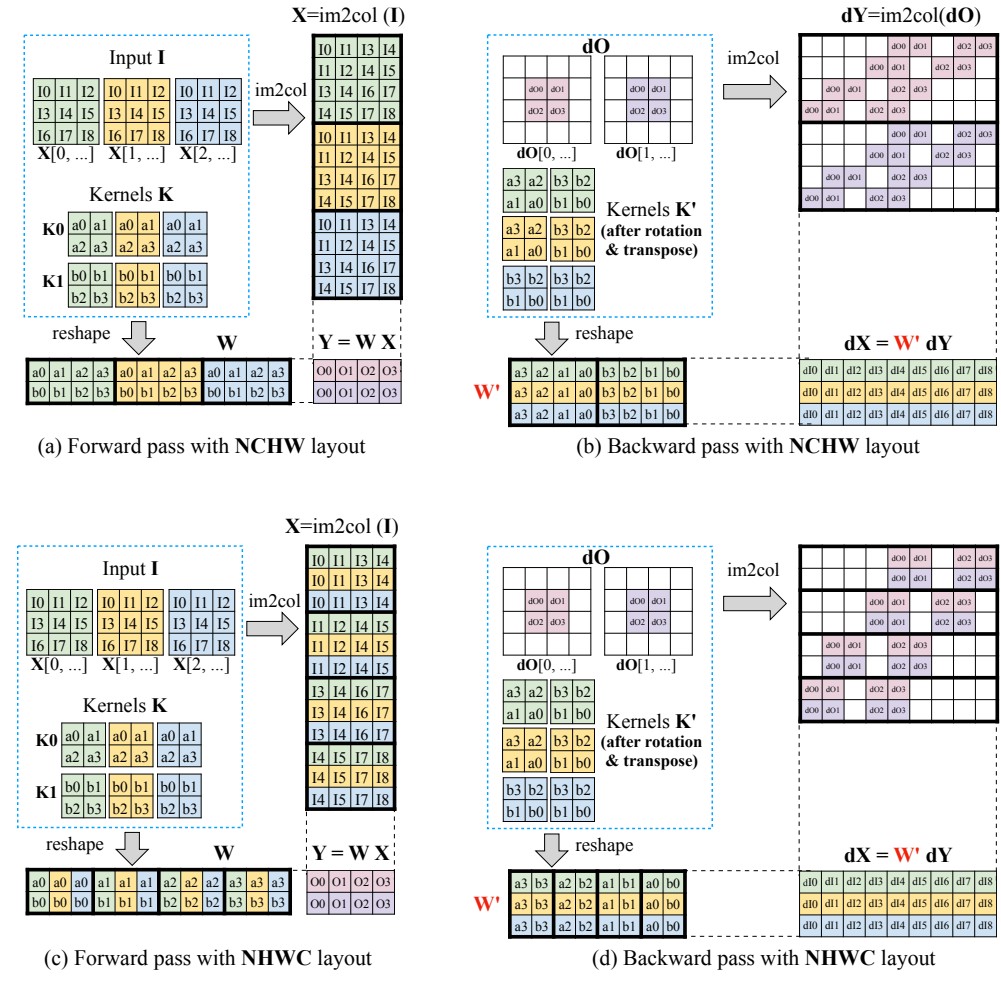

Figure 8: Illustration of GeMM with NCHW and NHWC format. Different colors represent different channels.

## B    N:M TRANSPOSABLE MASK IN THE BACKWARD OF CNN

As suggested in (Zhou et al., 2021), the N:M sparsity The N:M transposable mask (Hubara et al., 2021) guarantees that both the weights matrix $\mathbf{W}$ and its transpose $\mathbf{W}^T$ follow the same sparsity

pattern. This design works well for linear layer, whose forward pass is calculated with $Y = WX$ and backward pass is calculated with $dX = W^T dY$. However, the implementation of CNN with GEMM is different. Specifically, we have $\mathbf{W} \in \mathbb{R}^{C_O \times (C_I K_h K_w)}$ for the weights matrix of CNN. In the backward pass, CNN flips each kernel first vertically and then horizontally and performs the kernel-wise transpose. Thus, as in Fig. 1(c), $\mathbf{W}^T \in \mathbb{R}^{(C_I K_h K_w) \times C_O}$ is different from the matrix $\mathbf{W}' \in \mathbb{R}^{C_I \times (C_O K_h K_w)}$ in the backward pass of CNN in most cases. Nevertheless, this N:M transposable mask can still accelerate the CNN backward under some conditions.

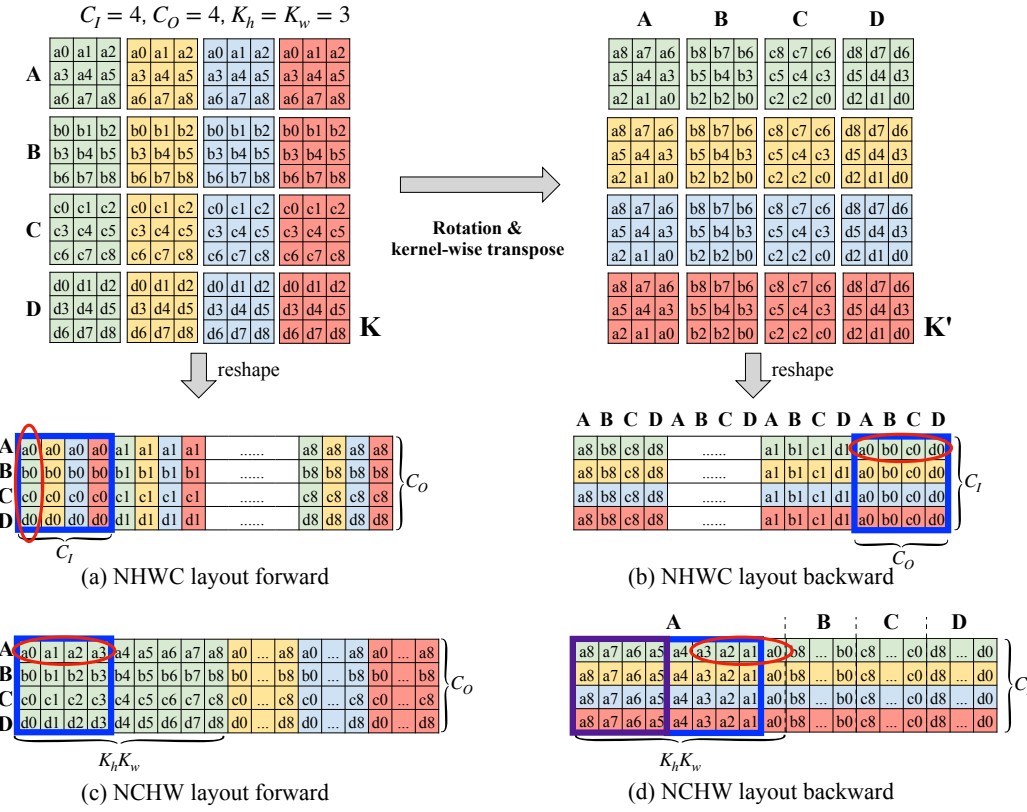

Figure 9: Comparison of NHWC and NCHW layouts of N:M mask in both forward and backward pass.

**Layout.** The first requirement is the layout of the weights in the memory. For example, suppose $N = 2$, $M = 4$, the size of the input channel is $C_I = 4$ and the size of the output channel is $C_O = 4$. Thus, we can divide the kernel matrix $\mathbf{W} \in \mathbb{R}^{4 \times (4 \times 3 \times 3)}$ into several $4 \times 4$ blocks. As shown in Fig. 9(a)(b), when we store kernels with NHWC format, we collect $a0$ with green color, $a0$ with orange color, $a0$ with blue color, and $a0$ with red color into a row of a block (*i.e.*, the blue box) for the forward pass, and this row meets the N:M constraint. Based on the N:M transposable mask, the column of $a0$, $b0$, $c0$, and $d0$ with green color also satisfies the N:M constraint. In the backward pass, this blue block can still be kept. Meanwhile, $a0$, $b0$, $c0$, and $d0$ with green color is collected as a row. Thus, the same sparse pattern can be maintained in the backward with N:M transposable design. When the weights is stored with NCHW format, as shown in Fig. 9(c)(d), the green cell $a0$, $a1$, $a2$, and $a3$ form a row of the blue block. Again, the column of $a0$, $b0$, $c0$, and $d0$ with green color also satisfies the N:M constraint. However, in the backward pass, the same block cannot be kept. Thus, the sparse pattern may not meet the N:M constraint. For example, in forward pass, green cells $a2$ and $a3$ are kept, and the green cells $a4$ and $a7$ are kept. In the backward, the green cell $a1$, $a2$, $a3$, and $a4$ form a new vector with the size $M = 4$, but this vector has three cells $a2$, $a3$, and $a4$ with mask "1". To this end, additional operations such as re-indexing and regrouping are required to make the weight matrix in the backward meet the N:M constraint. In summary, the N:M transposable mask can accelerate the CNN backward pass with NHWC layout GEMM implementation, but it cannot be directly applied to the NCHW layout.

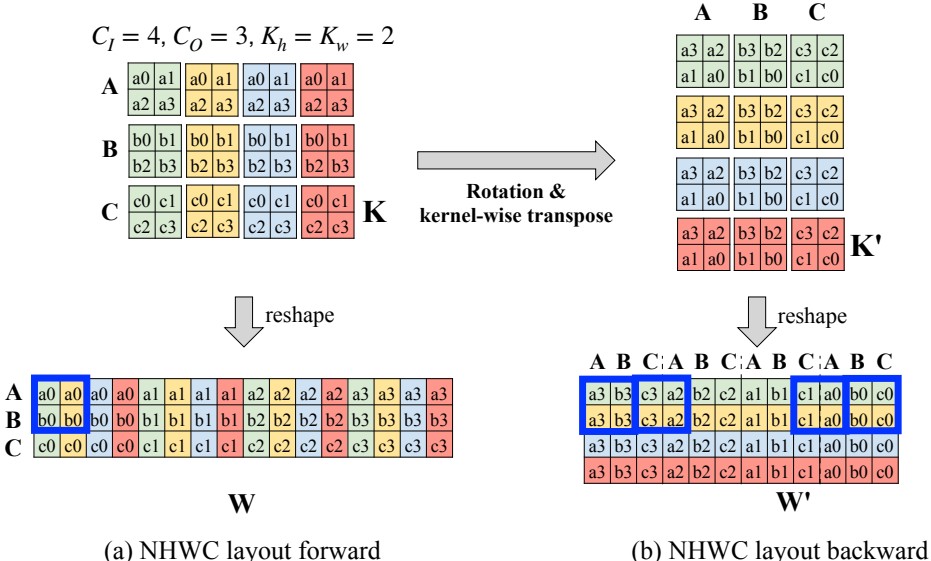

Figure 10: Example of N:M transposable mask when $C_O \mod M \neq 0$.

**Shape.** Moreover, even if the weights are arranged in `NHWC` layout, we show some cases that the N:M transposable mask may not bring the acceleration on backward as well. Considering the example with $C_O = 3$ and $M = 2$, as shown in Fig. 10, we collect $a0$ with green color, $a0$ with orange color into a row of a $2 \times 2$ block (*i.e.*, the blue box) for the forward pass, and this row meets the N:M constraint. Based on the N:M transposable mask, the column of $a0$, $b0$ with green color also satisfies the N:M constraint. However, in the backward, the $a0$ is grouped with $c1$, and the $b0$ is grouped with $c0$. Thus, the same sparsity pattern cannot be guaranteed. To this end, the output channel $C_O$ should be divisible by $M$, *i.e.*, $C_O \mod M = 0$.

## C   ALGORITHM SUMMARY OF HRBP/HRBP++

In this section, we summarize the detailed implementation of HRBP/HRBP++ below:

---

**Algorithm 1:** HRBP/ HRBP++

---

**Input:** unstructured mask $m$, density ratio of mask $r$, number of clusters $t$, density ratio of kernel $s$, number of input channel $C_I$, the number of output channel $C_O$, the width of the kernel $K_w$, the height of the kernel $K_h$, and the minimum number of rows of each group $B_1$

**Output:** Dense groups

Obtain kernel-wise mask $m'$ based on the number of non-zero cells within each kernel of $m$;

Divide the rows in $m'$ into $t$ equal-shape groups $\{g_1, g_2, ..., g_t\}$ with hypergraph partitioning;

**for** $g_i \in \{g_1, g_2, ..., g_t\}$ **do**

    **if** $g_i$ has no less than $B_1$ rows **then**

        Sort columns of $g_i$ from high to low based on the number of non-zero cells of each column;

        Select the $\frac{r}{s}C_I$ columns with the most number of non-zero;

        Extract the corresponding kernels $\mathcal{K}$ based on selected columns in $g_i$;

        **if** $s < 1$ **then**

            Count the non-zero weights of each cell based on all kernels in $\mathcal{K}$; **// HRBP++**

            Select $sK_hK_w$ cells with the most number of non-zero weights as the pattern $P$;

            Apply pattern $P$ to all kernels $\mathcal{K}$ and output them as a dense group;

        **else**

            Output all kernels $\mathcal{K}$ with all $K_hK_w$ cells as a dense group; **// HRBP**

        **end if**

    **end if**

**end for**

---

# D  NOTATIONS

In this section, we summarize all mathematical notations in this paper to help understand the motivation and the operation clearly.

Table 4: Summary of notations in this paper.

| Notation | Range | Definition | Equation |
|---|---|---|---|
| $C_I$ | $\mathbb{N}+$ | Number of input channels | |
| $C_O$ | $\mathbb{N}+$ | Number of output channels | |
| $K_h$ | $\mathbb{N}+$ | Kernel height | |
| $K_w$ | $\mathbb{N}+$ | Kernel width | |
| $H_I$ | $\mathbb{N}+$ | Height of input 2D feature map | |
| $W_I$ | $\mathbb{N}+$ | Width of input 2D feature map | |
| $H_O$ | $\mathbb{N}+$ | Height of output 2D feature map | |
| $W_O$ | $\mathbb{N}+$ | Width of output 2D feature map | |
| $C_I'$ | $\mathbb{N}+$ | Number of input channels in simplicity form | $C_I' = C_I \times K_h \times K_w$ |
| $\mathbf{K}$ | $\mathbb{R}^{C_O \times C_I \times K_h \times K_w}$ | Weights of a 2D convolutional layer | |
| $\mathbf{K}_c$ | $\mathbb{R}^{C_I \times K_h \times K_w}$ | One filter | |
| $\mathbf{K}'$ | $\mathbb{R}^{C_I \times C_O \times K_w \times K_h}$ | 2D convolutional weights in backward pass | |
| $\mathbf{I}$ | $\mathbb{R}^{C_I \times H_I \times W_I}$ | 2D input feature map | |
| $\mathbf{O}$ | $\mathbb{R}^{C_O \times H_O \times W_O}$ | 2D output feature map | |
| $\mathbf{dI}$ | $\mathbb{R}^{C_I \times H_I \times W_I}$ | Gradients w.r.t. the 2D input feature map | |
| $\mathbf{dO}$ | $\mathbb{R}^{C_O \times H_O \times W_O}$ | Gradients w.r.t. the 2D output feature map | |
| $\mathbf{dK}$ | $\mathbb{R}^{C_O \times C_I \times K_h \times K_w}$ | Gradients w.r.t. the kernels | |
| $\mathbf{W}$ | $\mathbb{R}^{C_O \times (C_I K_h K_w)}$ | Convolutional weights in GeMM (weights matrix) | $\mathbf{W} = \texttt{reshape}(\mathbf{K})$ |
| $\mathbf{W^T}$ | $\mathbb{R}^{(C_I K_h K_w) \times C_O}$ | Transpose of weights matrix | |
| $\mathbf{W}'$ | $\mathbb{R}^{C_I \times (C_O K_h K_w)}$ | Convolutional weights of backward pass in GeMM | $\mathbf{W}' = \texttt{reshape}(\mathbf{K}')$ |
| $\mathbf{X}$ | $\mathbb{R}^{(C_I K_h K_w) \times (H_O W_O)}$ | Input feature map in GeMM | $\mathbf{X} = \texttt{im2col}(\mathbf{I})$ |
| $\mathbf{Y}$ | $\mathbb{R}^{C_O \times (H_O W_O)}$ | Output feature map in GeMM | $\mathbf{Y} = \mathbf{WX}$ |
| $\mathbf{dY}$ | $\mathbb{R}^{(C_O K_h K_w) \times (H_I W_I)}$ | Gradients w.r.t. the output feature map in GeMM | $\mathbf{dY} = \texttt{im2col}(\mathbf{dO})$ |
| $\mathbf{dW}$ | $\mathbb{R}^{C_O \times H_O \times W_O}$ | Gradients w.r.t. the Convolutional weights in GeMM | $\mathbf{dW} = \mathbf{dY} \cdot \mathbf{X^T}$ |
| $\mathbf{dX}$ | $\mathbb{R}^{C_I \times (H_I W_I)}$ | Gradients w.r.t. the input feature map in GeMM | $\mathbf{dX} = \mathbf{W}' \cdot \mathbf{dY}$ |
| $m$ | $\mathbb{R}^{C_O \times (C_I K_h K_w)}$ | the binary mask of weights matrix $\mathbf{W}$ | |
| $m'$ | $\mathbb{R}^{C_O \times C_I}$ | kernel-wise mask | |
| $r$ | $[0, 1]$ | dense ratio of one convolutional kernels | |
| $t$ | $\mathbb{N}+$ | number of groups (dense blocks) | |
| $s$ | $[r, 1] \cap \{i/(K_h K_w)\}_{i=1}^{K_h K_w}$ | dense ratio of each kernel with size $K_h K_w$ | |

# E  MATHEMATICAL ANALYSIS OF CNN BACKWARD PROPAGATION

In this section, we give a detailed mathematical analysis of the CNN backward pass to show the reason for the transformation in the calculation of gradient w.r.t. the input. For simplicity, we assume that the number of input channels $C_I$ is 1, the number of output channels $C_O$ is 1, the stride is 1, and there is no padding operation in the forward pass. Thus, given the input $X \in \mathbb{R}^{H_I \times W_I}$ and the filter $K \in \mathbb{R}^{K_h \times K_w}$, in the forward pass of CNN, the output $O \in \mathbb{R}^{H_O \times W_O}$ is calculated by:

$$\mathbf{O}_{i,j} = \sum_{m=1}^{K_h} \sum_{n=1}^{K_w} K_{m,n} X_{i+m,j+n}, \tag{1}$$

where $1 \leq i \leq H_O$ and $1 \leq j \leq W_O$. As there is no padding operation and the stride is 1, we have $H_O = H_I - K_h + 1$ and $W_O = W_I - K_w + 1$. In the backward pass, given the gradients of the output $\dfrac{\partial L}{\partial O}$, based on the chain rule, the gradient w.r.t. each input pixel is calculated by:

$$\frac{\partial L}{\partial X_{a,b}} = \sum_{i=1}^{H_O} \sum_{j=1}^{W_O} \frac{\partial L}{\partial O_{i,j}} \frac{\partial O_{i,j}}{\partial X_{a,b}}, \tag{2}$$

where $1 \leq a \leq H_I$ and $1 \leq b \leq W_I$, and $\dfrac{\partial O_{i,j}}{\partial X_{a,b}}$ can be obtained by taking the difference on both sides of Equation 1. Thus, when input pixel $X_{a,b}$ contributes to the output pixel $O_{i,j}$, $\dfrac{\partial L}{\partial O_{i,j}}$ contributes to the gradient of $\dfrac{\partial L}{\partial X_{a,b}}$. In practice, Equation 2 can be represented as a full convolution between a 180-degree rotated filter $\mathbf{K}'$ and the gradient on the output.

Considering a specific example where the input has shape $H_I = W_I = 3$, and the kernel has size $K_h = K_w = 2$, as shown in Figure 11. For simplicity, we set $CI = C_O = 1$. In the forward, based on Equation 1, we have:

$$O_0 = I_0 * a_0 + I_1 * a_1 + I_3 * a_2 + I_4 * a3$$

$$O_1 = I_1 * a_0 + I_2 * a_1 + I_4 * a_2 + I_5 * a3$$

$$O_2 = I_3 * a_0 + I_4 * a_1 + I_6 * a_2 + I_7 * a3$$

$$O_3 = I_4 * a_0 + I_5 * a_1 + I_7 * a_2 + I_8 * a3$$

Take $O_0$ as an example, the gradient w.r.t. $I_0$, $I_1$, $I_3$, and $I_4$ are $\dfrac{\partial O_0}{\partial I_0} = a_0$, $\dfrac{\partial O_0}{\partial I_1} = a_1$, $\dfrac{\partial O_0}{\partial I_3} = a_2$, and $\dfrac{\partial O_0}{\partial I_3} = a_3$ respectively. Based on Equation 2, we can obtain the gradient of each input pixel:

$$\frac{\partial L}{\partial I_0} = \frac{\partial L}{\partial O_0} * a_0$$

$$\frac{\partial L}{\partial I_1} = \frac{\partial L}{\partial O_0} * a_1 + \frac{\partial L}{\partial O_1} * a_0$$

$$\frac{\partial L}{\partial I_2} = \frac{\partial L}{\partial O_1} * a_1$$

$$\frac{\partial L}{\partial I_3} = \frac{\partial L}{\partial O_0} * a_2 + \frac{\partial L}{\partial O_2} * a_0$$

$$\frac{\partial L}{\partial I_4} = \frac{\partial L}{\partial O_0} * a_3 + \frac{\partial L}{\partial O_1} * a_2 + \frac{\partial L}{\partial O_2} * a_1 + \frac{\partial L}{\partial O_3} * a_0$$

$$\frac{\partial L}{\partial I_5} = \frac{\partial L}{\partial O_1} * a_3 + \frac{\partial L}{\partial O_3} * a_1$$

$$\frac{\partial L}{\partial I_6} = \frac{\partial L}{\partial O_2} * a_2$$

$$\frac{\partial L}{\partial I_7} = \frac{\partial L}{\partial O_2} * a_3 + \frac{\partial L}{\partial O_3} * a_2$$

$$\frac{\partial L}{\partial I_8} = \frac{\partial L}{\partial O_3} * a_3$$

As in Figure 11, we can perform a convolution operation between the 180-degree rotated kernels and the gradient w.r.t output $\dfrac{\partial L}{\partial O}$ with zero padding. Note that, zero padding on the output is necessary as we need to ensure the product of this full convolution has the same shape as the input.

## F  RESULTS ON TRANSFORMERS

Recently, the ViT (Dosovitskiy et al., 2021) shows that transformers (Vaswani et al., 2017) also play an important role in computer vision tasks. The transformer encoder contains a self-attention module and an MLP layer. Given an input sequence $\tilde{\mathbf{X}}$, the self-attention applies three linear transformations

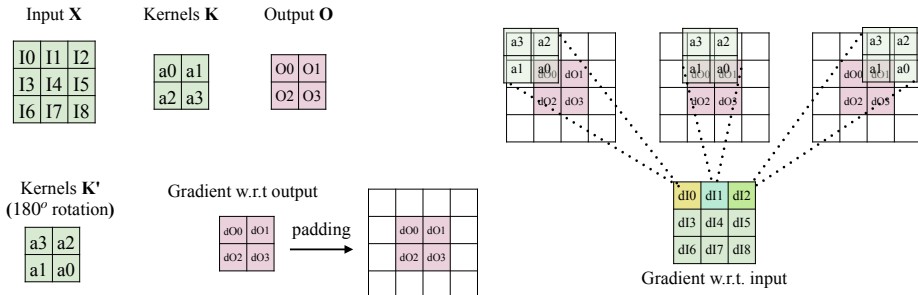

Figure 11: Illustration of the full convolution between the rotated kernels $\mathbf{K}'$. For simplicity, we use $dO$ to represent $\dfrac{\partial L}{\partial O}$.

to obtain the query $\tilde{\mathbf{Q}} = \tilde{\mathbf{W}}_{\mathbf{q}}\tilde{\mathbf{X}}$, the key $\tilde{\mathbf{K}} = \tilde{\mathbf{W}}_{\mathbf{k}}\tilde{\mathbf{X}}$, and the value $\tilde{\mathbf{V}} = \tilde{\mathbf{W}}_{\mathbf{v}}\tilde{\mathbf{X}}$, respectively. Then the output is obtained by $\tilde{\mathbf{Y}} = \text{Softmax}(\dfrac{\tilde{\mathbf{Q}}\tilde{\mathbf{K}}^{T}}{\sqrt{d}})\tilde{\mathbf{V}}$, where $d$ is the embedding dimension. Thus, all calculations of transformers with learnable weights are linear projections. To this end, the general form $\mathbf{dX} = \mathbf{W}^{T}\mathbf{dY}$ for the gradient w.r.t. the input is suitable for transformers in the backward pass and we can keep the same dense blocks at both forward and backward pass for sparse training of transformers.

Although the transformer does not encoder the shape transformation issues like CNN, we further apply our method on DeiT-Tiny (Dosovitskiy et al., 2021; Touvron et al., 2021) to show that the dense-block-based method is also effective in the training of transformers. The dense model of DeiT-Tiny can achieve 72.2% accuracy on ImageNet. With HRBP, we can achieve 72.7% accuracy with 60% sparsity.

## G    COMPARISON WITH THE GROUP LASSO PENALTY

The DessiLBI (Fu et al., 2020; 2022) proposes to use the group lasso penalty on each convolutional filter to prune less important filters. When first training a dense network, DessiLBI can find effective subnetworks (*i.e.*, "winning tickets" Frankle & Carbin (2019)) at different epochs of the dense training (*i.e.*, early stopping). By retraining from scratch, the structural sparsity can achieve comparable performance as dense networks. We follow the setting in DessiLBI and apply HRBP++ to VGG-16 on CIFAR-10. We compare DessiLBI with Lasso since baseline DessiLBI with group lasso penalty is ineffective at extreme sparsity level. The results are shown in Figure 12. Our method can also achieve comparable or even better accuracy than DessiLBl at the same sparsity level. Note that, DessiLBI follows the LTH setting which still needs dense training at the first step, while our method directly trains the sparse subnetwork from scratch without dense training.

## H    ABLATION STUDIES ON INITIALIZATIONS.

We by default apply our method to random unstructured sparse mask with ERK sparse ratio distribution (Evci et al., 2020) of each layer. In this ablation study, we explore the effect of different types of unstructured sparse mask initialization. In detail, we compare the default setting with 1) Random unstructured sparse mask with uniform sparse ratio distribution. 2) Unstructured mask from SNIP (Lee et al., 2019). 3) Unstructured mask from GraSP (Wang et al., 2020). We then apply HRBP++ to these unstructured masks and show the results in Figure 13. Noticeably, with SNIP and GraSP masks, HRBP++ can also achieve comparable accuracy at the same sparsity level. With uniform sparsity, the accuracy is slightly lower. This is consistent with the conclusion in (Evci et al., 2020) that ERK sparsity is beneficial. Thus, HRBP++ is robust with different types of mask initialization.

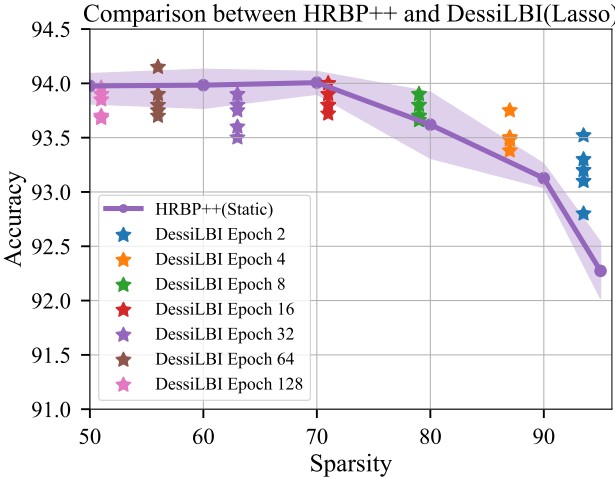

Figure 12: Comparison between HRBP and DessiLBI.

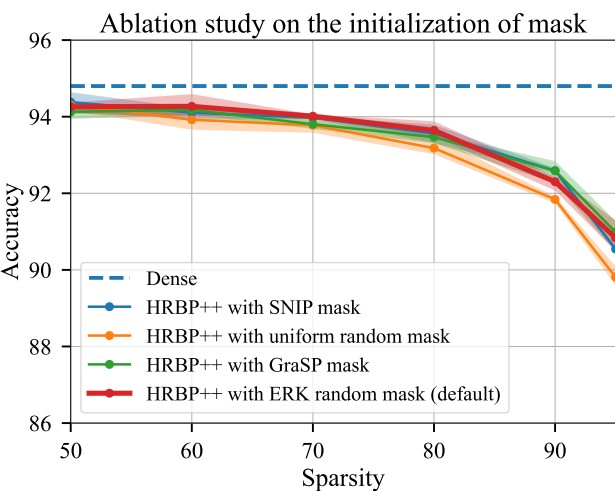

Figure 13: Ablation study with different types of mask initialization of ResNet-32 on CIFAR-10.

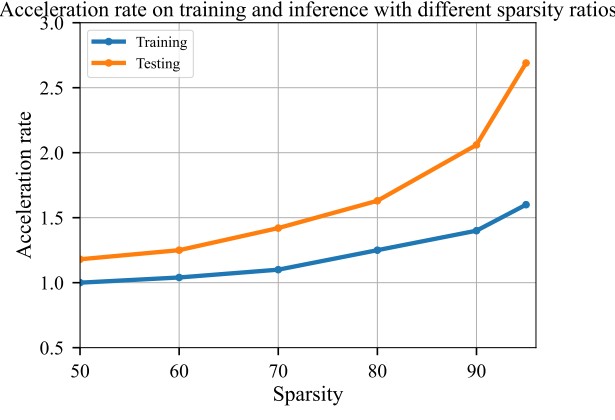

Figure 14: Training and inference acceleration on hardware of ResNet-32 on CIFAR-10.

