# OpenReview forum: "HRBP: Hardware-friendly Regrouping towards Block-wise Pruning for Sparse Training"
_ICLR.cc/2023/Conference — Submitted to ICLR 2023_

### Official Review · Reviewer_5xYn · 2022-10-23

**Confidence:** 2
**Clarity, Quality, Novelty And Reproducibility:** There are some confusing points, plea…
**Correctness:** 2
**Technical Novelty And Significance:** 3
**Empirical Novelty And Significance:** 3
**Recommendation:** 5

**Details Of Ethics Concerns:**

N.A.

**Strength And Weaknesses:**

Weaknesses:
1. From my point of view, the most important matter of the submission is whether $dW=dY  X^\top$ can be applied in backward directly. Though it is claimed and visualized in the content (for example, Figure 1) that the resulting $W^{'}$ is not $W^\top$, I have conducted gradient derivation by manually multiplying gradient of output and activation and my experience implied that dW can be arrived by matrix multiplication between dY and X. My past work may not be fully solid. However, one very obvious question will be: why padding is necessary in backward ? In another words, I am still confused about the submission's foundational motivation.

**Summary Of The Paper:**

This paper first claimed that element arrangement in forward's feature/parameter multiplication is different from the counterpart in backward, which leads to the inefficiency in previous sparse training acceleration method. Based on this observation, the paper proposed a block grouping method to aggregate non-zero elements in feature/parameter.

**Summary Of The Review:**

The submission take effort to visualize the idea and make itself understood but it fails on me. Hope I can be clarified in the following discussion.

---

> ### Author Response · Authors · 2022-11-16
> **The matrix multiplication in the backward and motivation of our paper**
>
> As discussed in Section 2.1, in backward propagation of CNN, we need to calculate both the gradient w.r.t the weights $\mathbf{dW}$ and the gradient w.r.t the input $\mathbf{dX}$.
> 1) For $\mathbf{dW}$, it can be obtained by  $\mathbf{dW}=\mathbf{dY}  \mathbf{X^T}$. Thus equation is satisfied for both linear layers and convolutional layers. We believe your previous experience can also support this conclusion.
> 2) For $\mathbf{dX}$, previous works simply apply $\mathbf{dX}= \mathbf{W^T} \mathbf{dY}$ to calculate it. This form is applicable to linear layers, but not applicable to CNN. As in Figure 1(b)(c), we need to conduct shape transformation on weights matrix W in backward, which obtains the gradient with $\mathbf{dX}= \mathbf{W'} \mathbf{dY}$ (See Section 2.1), rather than $\mathbf{dX}= \mathbf{W^T} \mathbf{dY}$. To this end, sparse patterns based on $\mathbf{W}$ or $\mathbf{W^T}$ from previous works cannot be kept on W’. Thus they cannot bring the expected acceleration on CNN backpropagation.
>
> In this paper, we focus on sparse CNN training with fine-grained structure pruning. In detail, we group the unstructured weights W into several dense blocks. Thus, the large matrix multiplication of sparse weights can be decomposed into several small dense matrix multiplications on hardware. With HRBP/HRBP++, dense blocks found on $\mathbf{W}$ can also be kept in $\mathbf{W'}$. Thus, our method can accelerate both the forward and backward propagation of sparse CNN training with fine-grained sparsity on hardware.

---

> ### Author Response · Authors · 2022-11-16
> **Why padding is necessary in backward pass**
>
> As in Figure 1(b), the padding operation is applied to the gradient of the outputs $\mathbf{dO}$.
> In practice, the gradient w.r.t. the input $\mathbf{X}$ is conducted by a full convolution between $\mathbf{dO}$ and the 180-degree rotation kernels $\mathbf{K'}$. Thus, the padding operation is necessary since we need to ensure the output from the full convolution can match the shape of the input $\mathbf{X}$. For more details, please refer to the mathematical analysis and the specific example in Appendix E.

---

### Official Review · Reviewer_fvk3 · 2022-10-25

**Confidence:** 5
**Correctness:** 3
**Technical Novelty And Significance:** 2
**Empirical Novelty And Significance:** 2
**Recommendation:** 5

**Clarity, Quality, Novelty And Reproducibility:**

(1) Can you provide some intuitions on whether the proposed form of sparsity is applicable to other neural network operations? DepthWise convolution? Self-Attention? etc.?

(2) Prior work studied other neural networks (recurrent, deeper resnets). Have you considered applying your method on these neural network to better show its generality? Or is this a limitation of your method that can only be applied to reasonably small-sized models?

(3) While it is good to use a broad terms for the paper title, however I would suggest the title of the paper to clearly focus on the target goals of the paper. The paper reads as it is a technique that is general enough for different NN operations, but as the method and results demonstrate, it mainly targets convolution operations.

(4) The results in Table 1 demonstrate that HRBP is seemingly more robust across multiple iterations of training (less variations) -- I agree that it may not be notable difference, but do you have any intuition why it is the case?

(5) The paper covers reasonably high sparsity regime (e.g. 90% and 95% sparsity). What are the benefits of your method for other sparsity ratios? An ablation study---similar to prior work which you have cited---would provide insights about the trade-offs between accuracy-speedup. This is critically important because not all neural networks can be sparsified to the degree that you have studied.




**Strength And Weaknesses:**

**Strength**

- Unifying sparsity for both forward and backward pass that enables better utilization of existing hardware platforms for acceleration.

- Results on few convolutional neural network shows promising results both in terms of accelerating training and preserving model accuracy, comparable to unstructured sparsity accuracy.

**Weaknesses**

- The paper lacks discussion on how the proposed form of sparsity can be extended or applied to other (convolution/non-convolution) operations.

- The paper did not cover (or discuss the limitations) other form of neural networks from prior work (recurrent neural network in SNIP or deeper ResNet models).

**Summary Of The Paper:**

The paper aims to accelerate the training workflow of convolution operators through sparsity. While recent work has explored structured/semi-unstructured sparsity to accelerate inference, this work targets training. The main challenge in leveraging sparsity in both forward and backward pass is crafting a sparsity mask that is transposable (or in the case of convolution the sparsity patterns are preserved after reforming the kernels). To achieve that, the paper suggests to apply sparsity mask in a kernel-wise manner, which arguable the main difference between this work and Rumi et al.). Leveraging this form of sparsity enables to preserve a hardware-friendly sparsity pattern for both forward and backward pass unlocking the opportunity to accelerate convolutional neural network training.

**Summary Of The Review:**

The paper tackles an interesting problem, accelerating sparse training. However, the domain and target workloads seem to be very narrow and not generally applicable to different neural network operations and workloads (or at least is not supported by the results). Even for the models covered in the paper, they are arguable old models and dataset, which still valuable, but raises the concerns of application of such method to a broad range of applications.

---

> ### Author Response · Authors · 2022-11-16
> **Apply our method on other networks to show the generality.**
>
> Although our HRBP and HRBP++ intend to solve the special issue of shape transformation in CNN backward pass, they can be extended to networks other than CNN as well, rather than limited to CNN or small-size models. For networks other than CNN, such as RNN and self-attention, the operation with learnable weights is usually linear projections. Thus, these networks don’t suffer from the shape transformation issue of CNN. In this case, the extracted dense blocks can be maintained in the backward pass naturally.
>
> Following your suggestion, we apply our method to deeper ResNet (i.e., ResNet-56 on CIFAR-10) in Section 4.5. As in Figure 7(c), our method is still effective in deeper nets at all sparsity levels. Moreover, we apply HRBP to transformers. Specifically, we choose DeiT-Tiny on ImageNet. As in Appendix F,  With HRBP, we can achieve 72.7% accuracy with 60% overall sparsity, while the accuracy of dense DeiT-Tiny is 72.2%. Thus, our method is also effective in networks other than CNN. We will add more results with transformers in the final version.

---

> > ### Comment · Reviewer_fvk3 · 2022-12-06
> > **Thanks for the rebuttal**
> >
> > Targeting only CNN models makes the scope of the paper relatively narrow. As other reviewers also highlighted, CNN models are relatively old with less number of applications. Adding results for additional transformer models would make your paper stronger and with a broader range of applications. Since your work targets training speedup, have you observed any speedup for transformer models? Did you sparsity both projection layers and self-attention calculation? Did you uniformly sparsify the attention layers?

---

> ### Author Response · Authors · 2022-11-16
> **Why HRBP is more robust across multiple iterations of training (less variations)?**
>
> Thanks for the interesting observation!  One possible reason is that HRBP/HRBP++ has less freedom on the mask (small mask diversity) than other pruning methods, as we have additional requirements for the sparse pattern of each kernel.

---

> ### Author Response · Authors · 2022-11-16
> **Ablation studies on other sparsity ratios.**
>
> Thanks for pointing out these important ablation studies. Following your valuable suggestion, we vary the sparsity from 50% to 95% to explore the trade-offs between accuracy and speedup. As shown in Figure 7 of Section 4.5 in the revision, our method can achieve comparable accuracy as unstructured pruning methods at all sparsity levels.

---

> > ### Comment · Reviewer_fvk3 · 2022-12-06
> > **Thanks for the rebuttal**
> >
> > It seems that your method is comparable to prior work in terms of accuracy with the exception of sparsity ratio above 80%-90% (Table 1 results also show this trend).
> >
> > Thanks for adding Figure 14, however, I have hoped that you added the results for deep models, as speedup is one of the main premise of your work. In Figure 14, can you have the breakdown between forward and backward pass? What causes the training phase to enjoy less speedup? I suspect that could be because the backward pass contributes more to the e2e runtime. I think adding the runtime breakdown could better explain this.

---

### Official Review · Reviewer_oxCu · 2022-11-02

**Confidence:** 4
**Correctness:** 3
**Technical Novelty And Significance:** 2
**Empirical Novelty And Significance:** 2
**Recommendation:** 5

**Clarity, Quality, Novelty And Reproducibility:**

Clarity and quality: The presentation of the proposed methods is clear, but the definitions of symbols are missing, such as the range of C_O, C_I, etc. It is better to summerize the methods into explicit algorithms. How is the propose method affect the network structure?

It would be better to put the implementation illustration on the appendix, and summerize the method into an algorithm format.

Novelty: It can be further illustrated. What is the improvement on compression rate? How is the accuracy?




**Strength And Weaknesses:**

Strength:
1. The network pruning is an important topic, and the author has given sufficient background introduction.
2. The paper illustrate the implementation flow chart of HRBP and HRBP++ clearly.

Weaknesses:
1. There are spelling errors in the paper.
2. The paper mix up the programming languages and the math notations. It makes the presented methods difficult to follow. The definitions of symbols and math notations are missing.
3. The advantages of using the presented methods are unclear. What is the compression rate by using the methods? How is the impact on the accuracy? What is the current state of the art?



**Summary Of The Paper:**

This paper presents the Hardware-friendly regrouping towards block-based pruning (HRBP) method, and the HRBP++ method. It tests the method on CIFAR-10, CIFAR-100 and ImageNet dataset.

**Summary Of The Review:**

This paper presents the Hardware-friendly regrouping towards block-based pruning (HRBP) method, and the HRBP++ method. It tests the method on CIFAR-10, CIFAR-100 and ImageNet dataset. The paper can be further improved by clarifying the notations, and clarifying the propose method impact on data structure, compression rate and accuracy. It is also better to illustrate the initialization impact on the proposed method.

---

> ### Author Response · Authors · 2022-11-16
> **The presentation of our paper**
>
> We sincerely thank you for your suggestions in the writing and presentation of our paper. We uploaded a revised version based on your valuable suggestions, including adding a summary of math notation in Appendix D, a summary of methods into explicit algorithm format in Appendix C, etc.

---

> ### Author Response · Authors · 2022-11-16
> **The improvement on compression rate and the impact on the accuracy.**
>
> The compression rate is equal to the sparsity ratio used in this paper. It can be calculated by 1 / (1 - sparsity). For example, a 90% sparsity in Table 1 is equal to a compression rate of 10x. For the impact on accuracy, we can achieve comparable accuracy at the same sparsity level as other state-of-the-art unstructured pruning methods such as GraSP. We further add experiments that vary the sparsity from 50% to 95% to explore the trade-off between accuracy and sparsity in Section 4.5. Similarly, our method can always match the accuracy of unstructured pruning at all sparsity levels but bring training acceleration on hardware.

---

> ### Author Response · Authors · 2022-11-16
> **The advantage of our method.**
>
> In practice, unstructured pruning methods can maintain accuracy at very high sparsity levels, but cannot bring acceleration on hardware because the irregular mask leads to poor data locality and low parallelism, while structured pruning can bring hardware acceleration but suffer significant accuracy drop with large sparsity.
>
> Our method can take the advantage of both unstructured pruning and structured pruning. In detail, we can almost match the accuracy of unstructured pruning methods at an extremely sparsity level (more than 90%), and bring training and inference acceleration as structured pruning on hardware. Note that, we do not intend to push the limit of the compression rate as in previous unstructured pruning methods. Instead, we aim to accelerate the sparse CNN training at both forward and backward propagation on hardware, while previous fine-grained structure pruning methods cannot accelerate the backward pass of CNN due to the shape transformation from W to W’ in CNN backpropagation (see Section 2.1 for more detail).

---

> ### Author Response · Authors · 2022-11-16
> **The initialization impact on our method.**
>
> Basically, in our paper, we run each experiment 3 times and report the average value with the variance. This can suggest the impact of initialization with different random seeds. We further add ablation studies with different types of unstructured mask initialization in Appendix H, including a random mask with uniform sparse ratio distribution, the mask from SNIP, and the mask from GraSP. Our method is robust with different types of initialization.

---

### Official Review · Reviewer_WyN2 · 2022-11-03

**Confidence:** 5
**Clarity, Quality, Novelty And Reproducibility:** the paper is clean, with limited nove…
**Correctness:** 4
**Technical Novelty And Significance:** 2
**Empirical Novelty And Significance:** 2
**Recommendation:** 5

**Strength And Weaknesses:**

Strength:

1, generally this paper is well written,
2, the presented method is interesting.

weakness:

1, I had some doubts whether the compared baselines are reasonable. Why not directly using the group lasso penalty on the each convolutional filters like [r1,r2]. This method should be the naive way of pruning the filters.
[r1]Exploring Structural Sparsity of Deep Networks via Inverse Scale Spaces, TPAMI 2022
[r2] DessiLBI: Exploring Structural Sparsity on Deep Network via Differential Inclusion Paths. ICML2020
These should be compared, as group lass is exploring the structure sparsity of CNN filters.
I think the method of these papers is pretty strong competitor to the proposed method in this paper.

2, Page 4, HRBP paragraph, the third line: "weights, which is obtained by counting Thus, the". It seems missing something here?

3, As transformers make very good performance in many task, is it possible to apply the algorithm to transformer?  CNNs are a bit out of date (less attractive).

4, As this paper advocates the hardware-friendly algorithm. It should give some more implementation details: is the HRBP/HRBP++ implemented by cuda? How's the implementation of baselines in Fig. 5? Is the comparison apple-to-apple? Are the authours using C++ or Python to run the baselines in Fig. 5?

5. Fig.6 should be results from HRBP++. However the caption refers to HRBP instead.
6. It would be better if the authors could have some discussion about the practical usage of the proposed method. It seems that it is not so worth it  having only 2 times faster speed while dropping almost all of the original parameters and suffering from significant performance drop.

**Summary Of The Paper:**

This paper studies the hardware-friendly regrouping in order to make structured sparsity. It aims at brining the training acceleration on GPU. The algorithm is conducted  on the kernel-wise mask. Some experiments are validated the show the efficacy.

**Summary Of The Review:**

Please clarify my questions about the weakness.

---

> ### Author Response · Authors · 2022-11-16
> **comparison with DessiLBI and whether the compared baselines are reasonable**
>
> Thanks for sharing these interesting papers. We’d like to highlight the following differences between the suggested papers and our work. We use DessiLBI to denote them for simplicity, since [r1] is the extension of [r2].
> Firstly, DessiLBI aims to explore structural sparsity, where the entire filter is pruned. While our paper on sparse training with fine-grained structured sparsity, by extracting a few dense blocks from unstructured sparse weights. Thus, they are two different types of pruning.
> Secondly. DessiLBI requires the dense training (with early stopping) at first to find a proper structural sparsity with well-defined criteria (“winning ticket”), and then retrain the subnetwork to match the accuracy of dense nets (Section 5.4 of [r2]), which is the setting of the Lottery Ticket Hypothesis (LTH). While our work aims to directly train a random subnetwork from scratch without any dense training (i.e., sparse training).
>
> To this end, it may be unfair to compare HRBP with DessiLBI directly. Thus, we consider other sparse training methods such as SNIP, GraSP, and random pruning as baselines in Table 1 for fair comparisons. Following your suggestion, we add the comparison with DessiLBI in Appendix G. In detail, we follow the setting in DessiLBI and run HRBP++ with VGG-16 on CIFAR-10. Our method can also achieve comparable or even better accuracy than DessiLBl at the same sparsity level without any dense training.

---

> ### Author Response · Authors · 2022-11-16
> **Apply our method to transformers**
>
> In transformers, the operation with learnable weights is linear projections. In this case, the general form $\mathbf{dX} = \mathbf{W}^T \mathbf{dY}$ for the gradient w.r.t. the input is suitable. Thus, the transformer does not suffer from the shape transformation issue in CNN backpropagation. The dense blocks extracted from the sparse weights can be maintained naturally. Thus, we can also obtain acceleration at both forward and backward passes on linear layers with HRBP.
>
> Following your suggestion, we further apply our method to DeiT-Tiny on ImageNet during this short rebuttal time. As in Appendix F,  With HRBP, we can achieve 72.7% accuracy with 60% overall sparsity, while the accuracy of dense DeiT-Tiny is 72.2%. Thus, we can still match the accuracy of dense networks and obtain acceleration. We will add more results with transformers in the final version.

---

> ### Author Response · Authors · 2022-11-16
> **Implementation details of our method on hardware**
>
> We did implement our method with CUDA and measure the training time acceleration on GPU.
> In detail, we rewrite the $\texttt{nn.Conv2d}$  module of Pytorch with CUDA to run CNN with sparse weights. Given the dense blocks found by HRBP/HRBP++, the original convolution computation will be decomposed into several computations corresponding to dense blocks, and these computations will be performed in parallel by threads.
> For baselines in Fig. 5, for a fair comparison, we implement networks using C++, which calls the cuDNN. And we measure both the forward and backward time of each convolutional layer on the same GPU.
> We add more implementation details in Section 3.3 of the revised version.

---

> ### Author Response · Authors · 2022-11-16
> **The practical usage of our method**
>
> Our work aims to train sparse CNN from scratch. The relatively large performance drop only happens at extremely sparsity levels (>90%). Sparse training with unstructured pruning methods such as SNIP, GraSP, and the LTH also have similar conclusions. We further add ablation studies with different sparse ratios to show the trade-off between accuracy and sparsity in Section 4.5 of the revised version. In practice, users can choose any level of sparsity to meet the accuracy requirements.

---

### Author Response · Authors · 2022-11-16
**General Response and Summary of Revisions**

We sincerely thank all reviewers for their insightful comments and constructive feedback on our submission. To address the concerns raised by the reviewers, we have added experiments and discussion in the revised manuscript, where the revisions are highlighted in blue. Here we summarize the major changes in the revised manuscript:

1. Presentation of our paper. We have fixed typos issues raised by Reviewer [WyN2] and [oxCu]. Meanwhile, we revise our title to focus on the convolution operation as suggested by Reviewer [fvk3]. Moreover, following the suggestions of Reviewer [oxCu], we add a summary of math notation and a summary of methods into explicit algorithm format to clearly illustrate our method.

2. As suggested by Reviewer [fvk3], we add ablation studies with different sparse ratios to show the trade-off between accuracy and sparsity in Section 4.5, which can also address the concerns of the compression rate issue from Reviewer [oxCu] and the practical usage issue from Reviewer [WyN2].

4. As suggested by Reviewer [fvk3], we add experiments with deeper nets such as ResNet-56 in Section 4.5.

5. As suggested by Reviewer [WyN2] and [fvk3], we add results on transformers to demonstrate the generalization ability of our method in Appendix F. Although transformer does not suffer from the shape transformation issue of CNN, our block-based method is also effective in the training of sparse transformers.

6. As suggested by Reviewer [WyN2], we add comparisons with DessiLBI on VGG-16 and discuss the major differences between them in Appendix G.

7. As suggested by Reviewer [oxCu], we add ablation studies on the initialization mask in Appendix H.

8. We add a detailed mathematical analysis in Appendix E to show the reason for shape transformation and padding operation in CNN backward pass, which can solve the concerns from Reviewer [5xYn].

---

### Decision · Program_Chairs · 2023-01-20

**Decision:**

Reject

**Justification For Why Not Higher Score:**

The reviewers unanimously suggest not to accept the current manuscript, so is the decision.

**Justification For Why Not Lower Score:**

N/A

**Metareview: Summary, Strengths And Weaknesses:**

This manuscript attacks the sparse training problem, i.e. how to train a sparse network from scratch. It presents the Hardware-friendly regrouping towards block-based pruning (HRBP) method, and the HRBP++ method. The method is evaluated by convolutional networks on CIFAR-10, CIFAR-100 and ImageNet dataset. However, all the reviewers have concerns about novelty, evaluation, and presentation. The reviewers unanimously suggest not to accept the current manuscript, so is the decision.